# Metaphysics, Universal Irony, and Richard Rorty's "We Ironists"

**Timo Airaksinen** 

Department of Practical Philosophy, University of Helsinki, FI-00014 Helsinki, Finland;
timo.airaksinen@helsinki.fi

**Abstract:** Richard Rorty speaks of "we ironists" who use irony as the primary tool in their scholarly work and life. We cannot approach irony in terms of truth, simply because, due to its ironies, the context no longer is metaphysical. This is *Rorty's challenge. Rorty's promise* focuses on top English Departments: they are hegemonic, they rule over the humanities, philosophy, and some social sciences using their superior method of ironizing dialectic. I refer to Hegel, Gerald Doherty's "pornographic" writings, and Gore Vidal's non-academic critique of academic literary criticism. My conclusion is that extensive use of irony is costly; an ironist must regulate her relevant ideas and speech acts—Hegel makes this clear. Irony is essentially confusing and contestable. Why would we want to use irony in a way that trumps metaphysics? Metaphysics, as defined by Rorty, is a problematic field, but irony can hardly replace it. At the same time, I admit that universal irony is possible, that is, everything can be seen in ironic light, or ironized. The purpose of this paper is to evaluate and criticize Rorty's idea of irony by using his own methodology, that is, ironic redescription. We can see the shallowness of his approach to irony by contextualizing it. This also dictates the style of the essay.

**Keywords:** sarcasm; dialectics; literary criticism; history; truth; Hegel; Doherty; Vidal



## 1. Modes and Methods of Irony and Sarcasm

We must distinguish between irony as a mode of being and a method.[1] An ironist like Richard Rorty uses irony as a rhetorical tool and linguistic trope that serves his plans, hence this study deals with the ironists' fortunes. They are existentially committed to irony and ironic attitudes—unlike an innocent user of occasional ironic speech acts. Their identity depends on their ironic attitude to life and social reality—they are *ironists*, or perhaps they use sarcasm so extensively that they look like skeptics, cynics, and nihilists.[2] Anyway, their existence is ironic *an sich*. Let us avoid too deep a dive into such murky classifications and discuss ironists in a wider sense that also includes pessimists and cynics and other devotees of sarcasm. They all use irony because it fits their overall understanding of the world and life—it comes to them as if naturally. Not all agents are ironists, Rorty calls these others metaphysicians (Rorty 1989, p. 74).[3] Certain people loathe irony and its related tropes and try to avoid the company of ironists. They still think in terms of truth, reality, and valid inference, and they want veridical evidence that supports their deliberations. They may even believe in moral truth, objective criteria of valuation, and logic of norms, that is, on "real essences." Therefore, they are metaphysicians, as Richard Rorty calls them in a semi-pejorative sense. In what follows, irony, as a method and mode of being, forms two separate categories, which is, I admit, a metaphysical theory devoid of any irony.

Yet, ironic expressions function independently of truth and therefore of metaphysics. Sometimes what is said is true, this is veridical irony (Dynel 2017), and truth is a metaphysical term. For example, you are paranoid, but it does not mean they wouldn't be after you, which says you are paranoid (true) and they may be after you (true), yet the sentence is deeply ironic. Compare: you say, what a fine car (false), when the car is junk

(true). Therefore, irony is independent of the truth and falsity of what is said. Verbal irony requires an intended audience to get off the ground. Metaphysical utterances also require an audience, of course, but their success does not depend on the audience's reaction. They are true and meaningful independently of it. However, the success of ironic utterances depends on an audience and their positive reaction. In a sense, an ironist can only speak to his fellow ironists. Others will misread the message, for instance, taking it as an insult. The finer aspects of Rorty's irony are esoteric—if metaphysics means common sense, as he says it does, it is exoteric (see Rorty 1989, p. 74). He must have in mind something else than philosophical metaphysics, which hardly is commonsensical, something like folk metaphysics—as cognitive scientists sometimes speak of folk psychology.

Rorty's philosophical approach is problematic but its rudimentary nature makes it difficult to criticize. However, by *contextualizing* it we can show how shallow and disappointing his approach is. This is what I do in this essay.

An ironist does not only watch irony emerging among his audience; on the contrary, he creates it in its bleak negativity. Irony is like violence that deforms its objects, ruins them, and lives out of such distorted effects. Hence, when we pursue the ever-elusive sources of irony we need to focus on the ironic attitude, which is to say that irony is more than a whimsical joke or a plain, neutral tool that we take out of our verbal toolbox and then put back. Obviously, this happens. But irony is a linguistic method, trope, and style, as well as an independent approach to the world and reality that it recreates. In this case, irony is not a miserable little verbal trick that may save the innocent speaker against an otherwise superior enemy; as such, it looks like a minor communicative vice—which is an ironic point about irony. But in Rorty's hands, irony becomes a major existential category. I ask, what is this existential irony like? Alas, an answer presupposes deafness to the ironic overtones of such an approach to irony. Think, for instance, of a serious and learned strategy to find the definition of irony and map its logical and linguistic properties, or to specify the necessary and sufficient conditions of its usage.[4] How to define something that is the very opposite of what is definite and definable?

Some texts are ironic per se, that is, they represent the textual version of *situational irony* (Airaksinen 2020a). Their ironies are embedded in the text so that the reader can find them there. She need not create them through her ironic method of reading, or do not represent *verbal irony* (Airaksinen 2020a; Bryant 2011). Here is an example, first in oratio recta: In Euripides' *Bacchants,* the god, Dionysus, pretends to accept king Pentheus' claim of having power over him—how could he? He is facing a god (Currie 2006). Their exchange proves that the king has no power at all—this is an example of situational irony and divine irony, or a god against man. Pentheus's supreme political power is worth nothing (Euripides, lines 610–20; Johnston 2017):

DIONYSUS: What punishment am I to suffer? What harsh penalties will you inflict?

PENTHEUS: First, I'll cut off this delicate hair of yours.

DIONYSUS: My hair is sacred. I grow it for the god.

The god pretends to ask the king about his fate as if the king had power over him. He does not, and hence the god's question is sarcasm based on his meiotic and cynical attitude towards earthly rules and rulers. The god rejects the king's values in a nihilistic manner; only his rites matter so that human life and death depend on celebrating them.

Next, sarcasm in oratio obliqua: Sade says that La Mettrie argues that conscience is your first executioner. Here M. de Sade utilizes a direct loan from Julien Offray de La Mettrie, who says conscience punishes you because of your wrongful actions and thus a penal justice system is superfluous (see Airaksinen 1995, p. 33). We regret what we did wrong, we feel guilty, we are ashamed, and we suffer. Therefore, we regret our crimes independently of lawful punishment. Sade transforms this into a sarcastic comment by condemning conscience that, for him, is essentially undesirable like any executioner is. In a crypto-Aristotelian fashion, you become an evil person by doing evil deeds, when you

repeat them long enough your conscience dies, and finally you are evil, free, and happy.[5] This idea approaches nihilism, that is, the denial of all value except egotistically understood freedom. Sade does not simply read La Mettrie sarcastically, he also explains his meaning in a new conceptual framework, that is, he moves from just restrictions to absolute personal freedom. This indeed is sarcasm because he praises the denial of value.

A note on sarcasm: Some people are ironists but not "sarcasts." Instead we find cynics and nihilists. I can make sarcasm my mode of being but that implies that I am a cynic. If I am an ironist, I am an ironist and perhaps nothing else, and that is all that can be said of me and my values. To be an ironist is a thin description, based on my linguistic attitudes and habits, unlike cynicism that is a thick description—it reveals much more about my valuations and values.[6] When I say I am an ironist I mean that I see irony everywhere and so I am prone to use irony as my primary tool. In the case of cynicism, I am prone to use sarcasm but much else is entailed as well, namely, valueless dispositions and blindness to all that is good and beautiful. Irony breaks language, but sarcasm breaks life and what value language talks about (Airaksinen 2020a, 2020b).

## 2. Rorty and the Policy of Universal Irony

I discussed above two literary examples and distinguished between irony and sarcasm. I used a descriptive method but I also presented reasons for my conclusions, which may sound metaphysical and as such unbecoming to an ironist. However, we cannot work without some distinctions and categories that we first explicate—scholarly work is different from tearoom chat. And all irony is derivative in the sense that it supervenes on facts. In this sense, it presupposes metaphysics and common sense. We should know the facts if we as ironists want to practice irony, and if we want to ironize irony, we better know some facts about irony. For an ironist, facts, logic, and reasoning may be useful, for a metaphysician they are all there is.[7] Let us then move over to the ironies of irony.

Sometimes irony is a speaker's free, subjective choice, as verbal irony. For an ironist, it is a perspective, attitude, and main language game. Irony everywhere. Sartre said that *angoisse* resides under your tram seat and Foucault found it in his mailbox. The same can be said of irony, it is under your seat and in the mailbox—when we read Sartre and Foucault ironically and bypass the problem of truth conditions. One may discuss whatever one likes in ironic terms, even if one must notice that sometimes this is easier and more natural, like my reading of Sade sand Euripides shows. Let us take an example from Socratic irony: Suppose I know a fact, but I pretend I do not know it and ask a member of my audience to explain it to me (see Currie 2006; Vasiliou 2013; Airaksinen 2020a). My discussants may refuse, of course, but it is too late; I already ironized the situation by using my favorite meiotic debating strategy. If the audience, except my victim, knows that I know, the situation also exemplifies dramatic irony: the audience knows more than the person who is on stage with me (see Goldie 2007; Williams 1992). The result is a complex combination of meiosis, verbal, Socratic, and dramatic irony, and, most importantly, I can generate this type of ironic situation at will, if I am willing to pretend that I do not know. In this way, the possibilities of irony are unlimited, and I mean both one's motivation and speech. The speaker can read any context in a suitably weird manner, for instance by using a *Verfremdungseffekt* (see Robinson 2008; Airaksinen 2021). Alienation looks prima facie ironic.

Rorty's *ironist* may remain sensitive to situational irony but he is not so much interested in *finding* it in the world than *creating* verbal irony as a ubiquitous linguistic trope –he is a universal ironist or ironist proper.[8] My question is, therefore, does this make sense? An affirmative answer implies both a consistently successful personal effort and the validity of the universal theory of irony. Suppose our ironist sees a car in the street. What can one do to transform this into ironic mentalese, or publicly ironize the relevant fact? One says, "what a glorious thing," and that does it. Can one train oneself in the black art of irony and sarcasm so that one transforms facts into verbal irony as if automatically? Could ironic mentalese be one's primary form of meta-cognition? When we discuss sarcasm,

we must ask whether anyone can be a thoroughgoing and consistent cynic and nihilist. Perhaps this problem is analogous to that of pessimism: can one see everything in a gloomy and negative light? Arthur Schopenhauer and M. de Sade tried hard (Dienstag 2009). It seems we must answer in the affirmative, even if we discuss a demanding and depressing position. A pessimist can always respond, say, with "yes, this is better but not good," or "but good won't last; soon all of it will be gone." Why not say a successful ironist is possible as well?

Rorty famously provides a universal method that transforms alleged facts into something else or ironizes them all.[9] He writes (Rorty 1989, p. 73):

> Ironists who are inclined to philosophize see the choice between vocabularies as made neither within a neutral and universal metavocabulary nor by an attempt to fight one's way past appearances to the real, but simply by playing the new off against the old. I call people of this sort "ironists" because their realization that *anything* can be made to look good or bad by being *redescribed*, and their renunciation of the attempt to formulate criteria of choice between final vocabularies, puts them in the position which Sartre called "meta-stable": never quite able to take themselves seriously. (My italics)

Perhaps no one can take such an ironist seriously. I cannot take Rorty seriously, as I now realize. He speaks of ironists but as well he might speak of conceptual relativists, cynics, anarchists, skeptics, and negative dialecticians.[10] He says, "I have defined 'dialectic' as the attempt to play off vocabularies against one another, rather than merely to infer propositions from one another, and thus as the partial substitution of redescription for inference" (Rorty 1989, p. 78).[11] As I read this, an ironist uses dialectic as a method of redescription to create an ironic effect, that is, a description in new and different terms or by offering a novel reading from an uncommon standpoint. Forget metaphysics, this is *Rorty's challenge*. He is not after the reasoned and logically grounded truth of a metaphysician; if he is not, his irony is its own goal and purpose as well as justification—no norms, rules, or valuations may apply to his ironizing process. Yet, some ironists are more interesting than others. Some are eminently successful like Rorty himself—this refers to his basic pragmatist position in philosophy.

Rorty says ironists happen, they practice literary criticism and presumably live and flourish mostly in the English Departments of certain top universities and they read texts and redescribe them, never taking themselves too seriously, that is, they play games for fun and hope for success. Their art is "the presiding intellectual discipline" (Rorty 1989, p. 83). They utilize the inexhaustible resources of the English language and English-speaking literature. The name of their game is "dialectic" because they play standard language against its ironized variants, and they do that consistently, effectively, and efficiently—ethics is not mentioned, perhaps because it is a metaphysical entity. The results accumulate, creating new conceptual schemes—or paradigms of language and interpretation—that in the long run overcome and replace the old and tired paradigms. Such English Departments are cultural and linguistic engines that influence and transform the social world and non-material culture, earning their members the status of leading theoreticians among the humanities and social sciences. This is *Rorty's promise*. It is a bold theory that I both accept and reject, dialectically of course.

Rorty's key idea, that is, the royal status of English Departments, was already developed by F. R. Leavis (1972) in his infamous treatise *Nor Shall My Sword* (1972). Denis Donoghue (1972) targets Leavis' views as follows—his anger is conspicuous:

> The declared aim is "to re-establish an educated, well-informed, responsible and influential public—a public that statesmen, administrators, editors and newspaper proprietors can respect and rely on as well as fear." In the nature of the case, such a public is bound to be a minority, but it would exert a moral influence far beyond its number. It would probably be found in the universities, more vigorously in York than in Cambridge, according to Dr. Leavis's evidence, but potentially in

any university which has made itself "a centre of human consciousness." This is the answer to "a present extremely urgent need of civilisation." Dr. Leavis does not offer to say precisely how such a university would be created, unless created, unless his insistence upon its "collaborative" and "creative" character is deemed to be enough. It would be the kind of university in which members of the English School would find it natural to discuss not only the great imaginative works of their own language but such exacting works as those of Whitehead, Collingwood, Michael Polanyi and Marjorie Grene.[12]

"There are already many universities in which Dr. Leavis' criteria are fulfilled," says the reviewer, just like Rorty: English departments will rule the intellectual elite environment that is the ideal university. In Rorty's case, the reason why this is so is as stunning as it is simple, in Leavis' case more complex; for Rorty, literature is "whatever literary critics criticize," or any text, and their working method is redescription or dialectical ironizing. How could this justify the supremacist claim? At least one must add, whatever texts they may *competently* criticize.

This talk about English departments is hegemonistic. Cultural hope and compassion live in the English departments all over the world. Are the top English departments the last strongholds and pinnacles of the ever-proud British colonialism and cultural imperialism, now of course perfected in the USA, in Hollywood, and by the major, global Anglo-Saxon publishing houses?[13] The local cultural life has less and less room to operate and flourish—is this not situationally ironic? In parks in Helsinki kids throw American pigskin, which indeed is a scary sight. The British Council sends their messengers all over the world to teach the English language and literature. An ironist may read this in hegemonistic terms, which again is a sarcastic comment on a cynically drawn picture.

Academics like to eulogize their work and workplace. One may read Gore Vidal as a counterpoint. In his collection of essays *Pink Triangle and Yellow Star* (1982), he directs his sarcastic gaze on academic literary criticism to deflate its narcissist claims. Authors may not appreciate academic critics, why should they? His evaluation of English departments is the opposite of Rorty's, and Leavis': he fails to see much promise of interesting redescription there. He does not want irony or any other type of redescription—he wants good prose. He says, "they flourish," but this ironically used metaphor is for him only a stylistic trick and as such of no theoretical or systematical interest. The only thing that matters is good prose—he writes about McDonaldization of academic literary criticism. At the same time, his essays provide a good example of the cost of irony and sarcasm. To write as he writes is to make bitter enemies, and of course, make some audiences laugh. Ironic writing can be both insulting and humorous, or humorous because it is so insulting—therefore it is so costly. Vidal writes (Vidal 1982b, pp. 146–47):

Unfortunately, the hardware store is pretty much all that there is to "literary" criticism in the United States. With a few fairly honorary exemptions, our academics write Brombertese [Victor Brombert, Princeton University], and they do so proudly. After all, no one has ever told them that it is not English. The fact that America's English Departments are manned by the second-rate is no great thing. The second-rate must live, too. But in most civilized countries second-rate are at least challenged by the first-rate. And score is kept in literary journals. But as McDonalds drives out good food, so these hacks of the Academe drive out good prose. At every level in our literary life they flourish.

However, when he writes "criticism" Vidal is sarcastic: "[Doris] Lessing's narrative devices are very elaborate. Apparently, the Canopian harmonious future resembles nothing so much as an English department that has somehow made an accommodation to share its 'facilities' with the Bureau of Indian Affairs" (Vidal 1982a, p. 114). He does not shy away from metaphysics: "without the idea of free will there can be no literature" (Vidal 1982a, p. 115). Here Vidal redescribes Lessing's creations using a caricature of English departments, which is to say Rorty's ironizing approach reflected back to where it originates.

Vidal ironizes the ironizers—Rorty should have done the same if he wanted to follow his favorite method. Anyway, scientists never respected the philosophy of science, and authors do not necessarily appreciate academic literary criticism and its hegemonistic claims. They ironize its claims just like academic criticism ironizes their target texts. If Rorty is right and criticism is typically ironic, it is only natural that so few people appreciate it, namely, redescription of texts in terms of ironic reading instead of illuminating them, say, via a display of the relevant metaphors, as Gerald Doherty does (see below).

## 3. Literary Criticism, an Example

As Rorty makes clear, ironists are not in the truth business and, therefore, they are disinterested in valid inferences, essential descriptions, and ethical principles, that is, metaphysics. Ironic redescriptions, in the name of dialectics, are what they do. Life without truth is another name for Rorty's challenge. Philosophers pursue the truth and nothing but the truth, I mean those poor metaphysicians. They may agree that truth is a tricky business, but could we live without it? When we see a proposition, can we always bypass its truth-claim and moral import and be content to redescribe it, thus solving all its problems like Alexander cut the Gordian Knot and thus solved its aporia.[14] Is it not ironic that mere redescription looks like a universal method of solving real problems? Redescription masks problems without solving them. Perhaps Rorty's favorite method is not the only possibility, or perhaps a critic can play with irony in some other way that reveals rather than masks? And in some cases, she will manage without ironizing the text.

An example follows. Does it apply and utilize Rorty's ironic dialectics of redescription? My answer is in the negative. Gerald Doherty's brilliant article "Imperialism and the Rhetoric of Sexuality in James Joyce's *Ulysses*" reveals the sexual and erotic aspects of the language of British colonialism: "From the start, British colonialism was thus also an erotic activity" (Doherty 1998, p. 208). Its key terms are penetration and conquest of fertile virgin territory, then possession and impregnation of it to make it carry fruit, or "the identification of a colonial poetic of a body with an erotic body." Here Doherty reads like a pornographer; he playfully calls his article pornographic (in personal communication), but he also seeks metaphors and descriptive idioms and discusses them—which is not quite what Rorty means. Does Doherty redescribe when he reads the original words and narratives in terms of sexual metaphors like virgin land, penetration, subjugation, and eventually rape?

This is to ask, is Doherty's reading ironic, as Rorty insists it could and, perhaps, should be? But he does not use the word irony, yet he redescribes his targets and thus sheds peculiar light on them. He calls his reading pornographic, and this may tempt us to conclude that his approach involves an ironic redescription. However, here irony does not follow from mere redescription; irony emerges because Doherty sees the target text as sexual and even pornographic, which it is not.[15] Here we need to refer again to that metaphysical chimera, truth—how could we avoid it? One may say that Doherty goes far beyond simple redescription: to call "penetration" a sexual term, not geographical or military, is not to use a novel vocabulary but to conflate two different contexts. He plays with certain linguistic ambiguities, and this is enough to create an ironic effect. Yet, his focus is on metaphors, and as we know, we have difficulties keeping metaphors and ironic expressions apart. Perhaps Rorty also should have said how to avoid this kind of confusion. To play with metaphors is certainly different from ironizing the topic, for instance, it can be said irony is always critical (Garmendia 2010). Doherty carefully avoids such an attitude—in this sense, he is not an ironist. His favorite trope is metaphor, but it can as well be metonymy.

What is the main difference between Doherty's and Rorty's use of irony? Rorty's ironists *create* verbal irony because to redescribe is to create a new interpretation. Doherty does not reinterpret. On the contrary, he pays attention to certain pre-existing linguistic tropes—and he *discovers* them and finds them surprising, amusing, and revealing. He does not redescribe anything. Instead he points out certain well-known metaphors, and then draws his conclusions on this basis. In other words, Rorty plays with verbal- and Doherty

with situational irony; therefore, Rorty masks a topic and Doherty reveals it. We may of course say that the combination of sex and colonialism sounds ironic, but this irony is closer to situational irony than verbal irony. Doherty does not create anything; he finds and reveals something that we have not seen before. He is an explorer, and that "something" contains its typical ironies. Therefore, Doherty is not an ironist in the sense of Rorty's "we ironists."

In his book, *Pathologies of Desire*, which is a study of James Joyce's *A Portrait of the Artist as a Young Man,* Doherty studies the sexual guilt feelings of Stephen Dedalus, noticing how guilt entails gaze: "Exteriorizing the place of the other, the gaze publicizes precisely those sexual acts Stephen has taken for granted were private" (Doherty 2008, p. 87). Here Doherty explicates the claim that guilt makes itself public via gaze, which entails an imaginary audience. In this way, guilt is related to shame in a lawlike manner. However, I do not find here much redescription via novel vocabularies. All this sounds like sound metaphysics, or good reasoning and psychological truth. The text here emulates psychological discovery. Why can we not say Doherty redescribes Stephen's feelings in psychological terms? Answer: he wants to provide a *true* description.

### 4. Hegel, History, and the Demise of Irony

It is difficult to agree with Rorty's and Leavis' idea of the hegemonic power of English departments, infinite resourcefulness of literature, unlimited competence of literary critics, and universality of redescriptive irony. I agree with them. Such superhero ironists may exist and I bow to their willingness and ability to find, create, and transmogrify ironies and touches of sarcasm for the benefit of the lesser mortals. However, academic History departments and their historians are good examples of the playing grounds—or killing fields—of ironist gamesters. History and histories are essential ironic arenas, as they deal with the "cunning of reason," as Hegel calls it, or the ironies of history.[16] Hegel plays with the dialectic of distributed vs. collective or universal ideas (Hegel 2004, p. 47):

> Particularity contends with its like, and some loss is involved in the issue. It is not the general idea that is implicated in opposition and combat, and that is exposed to danger. It remains in the background, untouched and uninjured. This may be called the cunning of reason—that it sets the passions to work for itself, while that which develops its existence through such impulsion pays the penalty and suffers loss. For it is phenomenal being that is so treated, and of this, part is of no value, part is positive and real. The particular is for the most part of too trifling value as compared with the general: individuals are sacrificed and abandoned. The Idea pays the penalty of determinate existence and of corruptibility, not from itself, but from the passions of individuals.

Historians' curse is their audiences' ubiquitous demand for truth—yet interpretation is foreign to the truth: no truth criteria for interpretative sentences may exist (see Ankersmit 2013). Therefore, they know that they only flirt with truth in bad faith, and therefore history is a field that a researcher, teacher, and audience will find difficult *not* to ironize. The main problem is truth claims; think of the ironic potential of the ever-so-amiable Stalin, the beginning and motivation of the Vietnam War, or the invention of concentration camps by the British during the Boer War. The audience wants the truth about them![17] A historian without a sense of irony is an oxymoron, unlike a literary critic who, anyway, is not targeting the truth. The historians' burden is heavy compared to literary critics. The former cannot afford to be a full-time gamester because they have such a hard time explaining the idea of historical truth to their amateur audiences.

Academic historians differ from literary critics in two respects: metaphysically speaking, literary critics only have a text that they read and comment on and that is all. Fiction represents nothing but its peculiar pseudo-reality—it operates in fictional possible worlds. Historians have texts and artifacts that they utilize to construct a world they must call true and real—it is a part of their game: behind the text is the hard reality of facts. Hence, as a profession, they cannot play the ironic game along with the literary critics. Yet, all historical

writing is deeply ironic in the situational sense: historians' writings mimic the past or mock it, and to mimic is to ironize by describing something in novel terms, often in a dialectical setting. Literary criticism is all metafiction and therefore not so serious business, unlike history that is at the same time ironic and metaphysical, serious business. Let us look at the following illustration (Cohn 1970, p. 119):

> From the very roots of Arthurian romance to the deeds of Maximilian I disguised in fictional form, the borderline between the romantic dream world and military reality was always indistinct: history was subject to the vagaries of imagination, and of the mind posed as sober truths.

We can read the last sentence in two ways. First, Arthurian romance and its chivalric fantasies are true in a special sense: they used to be part of the contemporary code of ethics. Second, they are not valid for us: their meaning is not personally binding to us. This is a distinction that literary criticism cannot draw; it is all imaginary. History has its real object in the past but literary criticism stays within the hermetic literary circle. History is unlike literary criticism; how could it be otherwise? Do historians think they are metaphysicians in the service of the truth? If they do not think like that, Rorty and Leavis can offer them their leadership in the field of humanities and a good slice of social sciences.[18] However, these academic fields are in the truth business.

## 5. The Costs of Irony

Verbal irony should destroy both metaphysics and our conventional contingencies. An ironist has a program for irony that occasional users of irony may not share. Rorty says irony is universal, ironists rule, and they are a critical, dialectical force (Owens 2000). I agree that the devil is everywhere simultaneously and it is polymorphic, and therefore I am fascinated by the ironists' ability to detect, create, and utilize irony whenever they look around and say something. The point is, we do not find ready-made ironies in our social environment (situational irony); on the contrary, we create it freely (verbal irony)—we are ironists, but most people are not. All this is related to the inherent costs of irony and especially sarcasm; we all know this cost, and it makes many or most people nervous.[19] They use irony sparingly. Many people are afraid of it. Some people never touch irony because they do not like or accept it; they do not want to see the world through tinted glass. They abhor being ironized. However, a true universalist argues that she can find irony everywhere and use sarcastic and ironist expressions in any social situation in a way that makes sense to her intended audiences, if not to all listeners.

Rorty the elitist says irony is not for all. Irony is private, irony is impotent, and "irony is the opposite of common sense." And "most people do not like to be redescribed," hence no irony for them—and this must apply to literary critics as well. "They want to be taken on their own terms" (Rorty 1989, p. 89). The worst part of irony is that it constantly turns against itself. Irony is an open-ended business, it is freely iterative, self-refuting, and forever non-conclusive. As such it is fascinating and valuable, but only for special purposes; on special occasions it is liberating, uplifting, and valuable. Irony seldom empowers the target, but it certainly empowers the speaker. Irony is, as Rorty says, derivative and post hoc: you need something to comment on, and therefore the Owl of Minerva is already flying when the ironist commences.

Here we have professor Rorty declaring "we ironists", which is an open invitation to ironize his standpoint (Rorty 1989, p. 79). For instance, President Kennedy in Berlin barked out, "Ich bin ein Berliner," and then "Laßt sie nach Berlin kommen" (26 July 1963). Rorty emulates him, does he not? This should not matter, according to an ironist, because by comparing the two great rhetoricians side by side we, anyway, ironize the context. I am sure Rorty wrote "we ironists" to be taken ironically; but ironical in which way, that is the next question? If he did not want to be ironic, he missed something, namely the peculiar situational irony in his own case, and such a failure is ironic, too. When I read Rorty, I think he should make it clear that to be an ironist is not all you can or should be. Irony may not be superfluous but it is a luxury.

Let us first examine the methods of freely ironizing your linguistic expressions and speech acts. We focus on the cost of irony and the ruined meanings—think of a sly smile during a funeral ritual. Call them internal and external costs.

> Where much of philosophy attempts to reconcile opposites into a larger positive project, Kierkegaard and others insist that irony—whether expressed in complex games of authorship or simple litotes—must, in Kierkegaard's words, "swallow its own stomach". Irony entails endless reflection and violent reversals and ensures incomprehensibility at the moment it compels speech. Similarly, among other literary critics, writer David Foster Wallace viewed the pervasiveness of ironic and other postmodern tropes as the cause of "great despair and stasis in U.S. culture, and that for aspiring fictionists [ironies] pose terrifically vexing problems.[20]

No one can be a free, universal, and self-consistent ironist; I mean no one can afford it. As Hegel says in his *Aesthetics*, and this applies also to those who write about art and literature: extensive use of irony creates negativity and shallowness of character. I may accept this as personal fate and destiny, but I cannot recommend it, which I do whenever I promote irony as my foremost method. Hegel writes, and this mature Hegel is no longer Rorty's favorite Hegel (Rorty 1989, pp. 78–79). Many Hegels may live peacefully together at the Philosophy Department, and this Hegel is in charge of the sheep dip (Monty Python). I quote from Hegel, my favorite Hegel (1975, Introduction: Irony):

> There are bad, useless people who cannot stick to their fixed and important aim but abandon it again and let it be destroyed in themselves. Irony loves this irony of loss of character. For true character implies, on the one hand, essentially worthy aims, and, on the other hand, a firm grip of such aims, so that the whole being of its individuality would be lost if the aims had to be given up and abandoned. This fixity and substantiality constitute the keynote of character. [ . . . ] But if irony is taken as the keynote of the representation, then the most inartistic of all principles is taken to be the principle of the work of art. For the result is to produce, in part, commonplace figures, in part, figures worthless and without bearing, since the substance of their being proves in them to be a nullity; in part, finally, there appear attached to them those yearnings and unresolved contradictions of the heart [ . . . ]. Such representations can awaken no genuine interest. For this reason, after all, on the part of irony there are steady complaints about the public's deficiency in profound sensibility, artistic insight, and genius because it does not understand this loftiness of irony, i.e., the public does not enjoy this mediocrity and what is partly wishy-washy, partly characterless. And it is a good thing that these worthless yearning natures do not please; it is a comfort that this insincerity and hypocrisy are not to people's liking and that on the contrary people want full and genuine interests as well as characters which remain true to their important intrinsic worth.

Hegel and Wallace agree. Big words and clever verbal games never completely mask thin contents; therefore, for Hegel, irony is a meiotic trope that we must use, if at all, with care. What does this say about us ironists? The problem any ironist must face, which makes her an unlikely creature, is the ever-escalating cost of actively and universally ironizing facts and values. I do not mean the external psychological cost of hurting and confusing people until they become aggressive and non-communicative. I mean a much deeper idea of intrinsic cost, namely, the various linguistic problems of meaning and sense ironic speech is vulnerable to. The listeners must always ask, first from themselves and then from others: What does she mean? How can I know? They know she does not mean what she says, but then, what does she mean? Irony entails pretense—which is an undesirable fact from Rorty's point of view.

We can distinguish between two types of audiences of verbal irony: intended listeners and non-intended general audiences. When I speak, I speak to someone; at the same

time, others may hear what I say. The level of understanding and acceptance of these audiences may differ considerably. In many cases, the unintended audiences may be the main problem because they have less background knowledge of the case and, hence, they cannot locate an ironic speech act in its proper context. The estimated costs of interpretation vary accordingly, and so do the audiences' psychological reactions. Of both, we can say that ubiquitous irony will be too much and perhaps intolerable in the long run, even in Rorty's simplified sense. Once you start redescribing redescriptions your time is up. Even where irony is a particular stylistic choice, it should not be used too extensively. When a member of the intended audience says, "I never know what you really mean," you see a clear warning sign and a piece of advice to start saying what you really mean, if you mean something—or to speak literally for a while. Put on your metaphysician's cap! Otherwise, the cost may be far too high, for example when you ironize your communication with an armed and very nervous robber. Metaphysicians live longer.

However, one may not feel compelled to answer an ironic quest; one may just shrug one's shoulders. Rorty may think that methodical applications of irony sharpen perception and illuminate the topic in a way that indeed can be called profound. Even a cynic may have an important story to tell, just like a skeptic does. In this way, behind the wrecked meanings, we find an outer layer of meanings that are somehow more expressive, evocative, and truer than the original meanings. Or we think that negativity as such is the key to the truth about the world—this is possible, I do not deny it. Perhaps Saussure was right: all meanings are created in contrast to other meanings; does not this mean that irony creates as well as destroys meanings?[21] Such contrasts can be ironized and yet they create meanings. Can ironized contrasts create new meanings? Can anything else create meanings? Are all meanings ironic? Before we answer, all this must be projected on the mirror of truth because it all depends on metaphysics. I cannot see how an ironist could bypass metaphysics.

## 6. Postscript: Rorty in Context

I have tried to show both why Rorty's idea of irony as dialectical redescription in terms of a novel conceptual framework may look so fascinating, novel, and creative in all its simplicity—and why its fails when his approach is *contextualized*. It may fail analytically, that is, his vocabulary does not do the work the way it should: Rorty's idea of metaphysics as common sense is odd and false. Anyone who reads a book on philosophical metaphysics can verify this. Yet we may redescribe metaphysics in the way Rorty does, and this is to say that he ironizes it—irony is redescription in novel terms, which is exactly what happens when he says metaphysics is common sense. We then lose the truth and many key ideas of linguistic meaning and principles of rationality. We are now free to read as we like, and write about what we see, feel, and experience. Yet, we gain much more: we can now play with the idea of universal verbal irony, that is, we can ironize whatever we want and hope that we have an audience that appreciates what we do. This looks like anarchistic daydreaming, but obviously the reading audiences have accepted the idea and kept on reading and buying his books.

For a critical reader, his writings create unique methodological problems. To read Rorty at the same time assuming a metaphysical point of view is unfair. He says he does not want metaphysics and his main idea is that we can manage without it. If we criticize him directly, or metaphysically, we fail to respect his basic intuitions. We should not do so, and yet we must criticize his main theses—how? We can ironize his ironic ideas, or as Hegel might have put it, we can take Rorty more seriously than he himself does. I do this by placing Rorty in context: his idea of irony looks simple, shallow, and rudimentary when we read it in the full context of irony, sarcasm, and their pragmatics. He does not discuss, say, situational irony, cost of irony, the essentially misleading effects verbal irony, and the distinction between metaphor and irony. His approach is too simple to be convincing.

We cannot read Rorty metaphysically, or relying on common sense, paying attention to truth and reality. Instead, we must take his texts as they are and apply his novel methodology to them. This idea dictates the style of this essay, which is free and as I hope

intuitive, without being common sense. Such indirect criticism is inconclusive; for instance, Rorty reads Hegel, so I read Hegel, but a different Hegel. I quote Gore Vidal, who has a low opinion of academic literary criticism. His sarcasm shows what he means by good prose, which he thinks is the essence of all worthwhile writing and reading—alas, for Rorty, no talk about essences makes sense. Gerald Doherty's innovative play with metaphors applied to some sexualized colonialist themes shows that we can redescribe without mentioning irony, or describe in various ways. Of course, we may refuse to draw a line between metaphor and irony, or irony and humor, but then we rapidly approach descriptive chaos. Philosophers typically want to keep such categories separate, but perhaps this is a vain hope: language tends to be a messy thing. But we should remember what Aristoteles (2013, chp. 3) says: "It is the mark of an educated man to look for precision in each class of things just so far as the nature of the subject admits; it is evidently equally foolish to accept probable reasoning from a mathematician and to demand from a rhetorician scientific proof." The criticism of Rorty should remain at the rhetorical level defined by his texts.

Rorty's methodology is not quite what it appears to be, as we see when we contextualize it. His emphasis on irony is superfluous, perhaps egregious. His rejection of metaphysics is unfounded and his eulogizing of the academic English Departments premature, and obviously not original; F. R. Leavis already said the same thing. My critical route is indirect and my arguments, for fairness's sake, are at the same conceptual level with Rorty's. I have avoided a "scholarly and scientific" approach because it is exactly what Rorty himself rejects so emphatically. Then the only fair approach is to ironize his idea of irony by redescribing it and its context once again in novel terms, as if dialectically. Such an approach has its problems but I do not see any other way when we check the existence of the Emperor's new clothes.

**Funding:** This research received no external funding.

**Institutional Review Board Statement:** Not applicable.

**Informed Consent Statement:** Not applicable.

**Acknowledgments:** Open access funding provided by University of Helsinki.

**Conflicts of Interest:** The author declares no conflict of interest.

## Notes

[1]  On the motivation, basic methodological approach, and stylistic choices of this essay, see *Postcript*.

[2]  See (Airaksinen 2020a, 2020b); also (Bernstein 2016; Kreuz 2020; Allen 2020; Gans 1997; Colebrook 2002; Lear 2011).

[3]  A metaphysician analyses old descriptions, he unlike ironists is still "attached to common sense": "The opposite of irony is common sense". See (Malachowski et al. 1990; Festenstein and Thompson 2001).

[4]  See (Bryant 2011). For instance, how to draw a line between metaphor and irony, if we do not want to mention the relevant intentions; see (Nathan 1992; Neuhaus 2016). Perhaps the best way to define irony is by indirectly distinguishing it from other, related linguistic tropes.

[5]  Sade's *Justine* tells a story of Thérèse, a young lady who never learn this truth. The book is cynical and amusing at the same time. Here is the key sarcasm: Socrates says only virtue brings about happiness, hence an evil and happy person is an oxymoron but yet recommended by Sade. See (de Sade 1967).

[6]  See (Väyrynen 2013; Williams 1985, p. 135f; Banasik-Jemielniak 2021; Camp 2012).

[7]  Wittgenstein's *Tractatus*, that paradigm of metaphysics in all its argumentative brilliance, ends with irony—that is why it is a great book. The ladder argument and the idea that what we cannot say we must keep silent of entails the existence of what is unknowable. This is derivative of Kant's idea that we cannot know das Ding *an sich*, yet he had so much to say about it. Metaphysics has its own ironies, or sarcasms, when its conclusions are skeptical and as such meiotic. See (Wittgenstein 1921, ## 6.54, and 7).

[8]  The problem with *finding* situational irony in the world is that one cannot say some situations are *in fact* ironic—this is a metaphysical proposition and truth claim. Irony, for Rorty, is a process of creation or verbal irony.

[9]  See e.g., (Inkpin 2013; Bacon 2017; Owens 2000).

10   See Feyerabend (1975) methodological anarchism in the philosophy of science: any evidence is good evidence, even dreams, omens, and magic. He signed published papers "Phantomas". His sarcasms were honest and calculated.—Rorty's project resembles Adorno's negative dialectics (see Brunkhorst 1988; Adorno 1981; Rorty 1989, pp. 56–57).

11   In his *Philosophy and the Mirror of Nature* (Rorty 1980, p. 18), Rorty is skeptical about "different vocabularies" and "alternative descriptions." The problem seems to be that they all talk about something that is independent of them—this is a realist intuition. Later he loses his faith on that "something" and becomes a kind of idealist.

12   See (Rorty 1989, p. 80; Moran 2010, p. 29ff, chp. "Leavis and the University"). For an example of such a reading, see (Detweiler 2016).

13   When Rorty eulogizes English departments, he becomes first deaf to linguistic relativity and the problems of translation and then guilty of cultural neo-colonialism. A non-English speaker cannot be an ironist without adopting the linguistic conventions of the native English speakers, which entails the hegemony of English speakers. The world in 1989 was not what it is in 2021. See (Loomba 2002, p. 94 ff). She discusses drunks and cannibals but dismsses exclusion like the translational impossibility of being an ironist: the definition of irony in English does not work well, say, in Finnish.

14   For Example, G. E. Moore asks, why is the sentence "Cat is on the mat but I do not believe it" paradoxical? We are invited to tell and not just cut through the knot by somehow ironizing the problem.

15   Doherty's irony is obvious. When watching porn, the person does not understand what he is seeing, he is stimulated but dumfound and glassy eyed, yet he cannot stop. He is irresistibly drawn in into the mystery that is the maelstrom of incomprehensible excitement. We have sexual porn but, in a generic sense of the term, also gardening, fitness, cars, fashion, and interior decoration porn. The victims read and watch something that is totally beyond them and their life experience. See (Nguyen and Williams 2020).

16   The Invisible Hand in economics (Adam Smith) is another good example of the essential (situational) ironies of a discipline, in this case economics; see (Bishop 1995).

17   Truth has three meanings: "true something", "truth about something", and "true sentence", or a quest for essences, an ultimate explanation, and the veridical description of a fact. Philosophers avoid the first and second and pretend that the third is the only possibility. We talk about true heroes, the truth about President Donald Trump's charisma, and the truth value of the sentences like "2 + 2 = 4" and "Moon is made of cheese".

18   We must not forget the movement called "Law as Literature". Laws are texts that the law-professions as ironists may read creatively, especially when they claim that laws apply without radical interpretation, think of the common-law system. The controversy is between "law is as it is written" and "law is as it is read". Lawyers do not want to play with such controversial ideas, yet law courts emanate all kinds of ironies. See (Ginsberg 1991; Duncan 1994).

19   For an opposite view, see (Kaplan and Winner 1995). Sometimes irony is humorous and as such prima facie innocent, see (Ruiz-Gurillo and Alvarado-Ortega 2013; Miller 2009).

20   *Wikipedia: Irony* (Wallace 1993, chp. 4.4). Available online: https://en.wikipedia.org/wiki/Irony#Irony_as_infinite,_absolute_negativity (accessed on 20 September 2021).

21   "It is only difference that defines meaning, not an ontological or verifiable linkage to extra-symbolic reality" (Alexander and Smith 1993, p. 157).

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
