# Peer review of "Metaphysics, Universal Irony, and Richard Rorty’s “We Ironists”"

_humanities, doi:10.3390/h10040106_

Round 1

Reviewer 1 Report

I admit that I was partly amused by some of the (intended) humorous accounts of the various ways irony operates in the dynamics of art and history, and the essay contains some scattered engaging and thought-provoking ideas and approaches that would be worthwhile to develop and follow-up. As evidenced, though, by the abstract, this article is mysterious and to me lacking the focus and clear line of argumentation (even if we do not consider proper objectives but a rather fluid essayistic approach to irony). It is a challenge for me to get a hold on the author’s line of argumentation and areas of investigation, except connecting irony to diverse concepts such as the irony of irony, ironists at English Departments, the art of redescription as a literary critique, the consequences of literature and history in terms of irony (and redescription). An attempt to set down the contours of this essay is made on page 3 (lines 79-88) but without successfully explaining the major aim of the essay. The dynamic use of subjects (he, she, it, they, I, we) (see for example page 4 lines 117-18) makes it more confusing, and also destabilizes the position of the author, which in any case seems to reliably on Rorty, referencing him much without really appropriate framing him in relation to the author’s intent. Many of the critics and thinkers involved are similar to the ones Rorty refers to in Contingency, Irony, and Solidarity. Further confusing is the use of irony as an entity of its own (page 10 – line 384). Add to this, a lack of proper block quoting and, as far as I can gauge, inconsistencies in the reference style at Humanities including a lack of a reference list in the end.

Saying all this, this is not in any way an uneven piece, just to me a very vague piece of writing which could be considerably focused and improved for better effect. I find that the discussions of Doherty and Vidal would benefit from a much more extensive ironic analysis. A caveat might be in what context it will operate? It might function more satisfactory as a companion piece to a special issue on irony, but then properly introduced as such.

Reviewer 2 Report

Richard Rorty can rightly be called the most modern philosopher, who called himself a philosophical ironic. His antipode in that sense is, for example, Martin Heidegger. The topic of philosophy using irony through Richard Rorty is current. The topic of the study can be evaluated positively.
Although the outline of the division of the study is good, it lacks three things within the division. It certainly lacks a conclusion that should be within the range that is common in studies. A study without scope simply cannot function in a scientific journal. It would also be asked to have a discussion to make it clear what the author was arguing with, how and on what basis he came to conclusions. What is inadmissible is the absence of final bibliographic records. There is definitely a need to add these three things to the study. The last subchapter has some features of discussion, but it would be good to draw up a separate subchapter discussion and a separate conclusion.
At the beginning of the study, the author seeks to grasp the essence of philosophical irony. It shows some specific jokes and sarcasms (for example La Metrie). It also explains the difference between ironism and sarcasm. Rorty's challenge is to get rid of metaphysics, which is one of the messages of his ironic approach. The author embarks on an evaluation of Rorty's approach. It deals with the academic and literary context of philosophical irony.
The author also analyzes Doherty's study, which deals with some aspects of James Joyce's work. I don't see any problem in this sequence. The reference to the connection between Doherty and Rorty should be deeper than the author does. I consider the connection between "cunning of reason" in Hegel and Rorty's irony to be a good observation by the author. This subchapter is definitely successful.
The text shows the knowledge of Hegel and Rorty on the part of the author (lexicon, allusions to various expressions of the mentioned philosophers). The text is of an academic nature. However, in order to publish it, the author must eliminate the shortcomings mentioned.

Round 2

Reviewer 1 Report

The through and careful revisions made by the author has contributed to an entirely different clarity in argumentation, focus and objectives and now we are reading a highly engaging and productive dialogue with Rorty. Especially the postscript frames this discussion in a reader-friendly capacity. Well done!

Author Response

thank you!

Reviewer 2 Report

I have commented on many aspects of the article in the last review, so I will not return to them. The author incorporated virtually all my arguments, remorse and recommendations into the new version of the article. The article can be considered an improved, reworked version. To a large extent, the article is of better quality because it is better structured, as well as because new moments have been added to it. The bibliography already exists in a separate section and is sufficient. The conclusion itself is referred to as "postscript". From my point of view, this can be seen as a conclusion to the study, I leave it to the editor to decide. I consider the study publishable, if possible accept the postscropt as a conclusion. 

Author Response

thank you!